# Cell atlases and the developmental foundations of the phenotype

**Alicia Lou**[1,2], **Mónica Chagoyen**[2¤a], **Juan F. Poyatos**[1¤b]*

**1** Logic of Genomic Systems Lab (CNB-CSIC), Madrid, Spain, **2** Computational Systems Biology group (CNB-CSIC), Madrid, Spain

¤a Current address: Centro de Neurociencias Cajal (CNC-CSIC), Alcalá de Henares, Spain
¤b Current address: National Museum of Natural Sciences (MNCN-CSIC), Madrid, Spain
* juanfpoyatos@csic.es

## Abstract

It is widely acknowledged that development shapes phenotypes, yet the extent to which genes with similar expression patterns during development lead to equivalent organismal phenotypes when mutated remains unclear. Here, we propose addressing this issue, which we term the $\mathcal{D}$evelopment–to–$\mathcal{P}$henotype, or $\mathcal{D}$–$\mathcal{P}$, rule, by leveraging single-cell gene expression atlases and phenotypic ontologies, using *Caenorhabditis elegans* as a model system. This framework quantifies the proportionality between developmental expression and phenotypic similarities, demonstrating that the relationship holds on average. Genes that strongly fulfill the rule exhibit broad "housekeeping" expression and are associated with systemic phenotypes, whereas weak similarities correspond to specific expression patterns and specialized phenotypes. Deviations from the $\mathcal{D}$–$\mathcal{P}$ rule provide insights into developmental divergence and phenotypic degeneracy, highlighting genes with narrow functional roles but systemic phenotypic impact. Furthermore, genes that closely adhere to the rule exhibit the highest pleiotropic impact on organismal traits. Our analysis also identifies cell types, such as ASK neurons, as key mediators of phenotype-specific gene contributions, exemplified by their association with chemosensory behavior and chemotaxis. These findings validate the $\mathcal{D}$–$\mathcal{P}$ rule and underscore the role of cells as critical mediators of the genotype-phenotype map, offering a unified framework to understand the developmental origins of phenotypic complexity.

## Author summary

Single-cell sequencing technologies now allow us to map gene activity during embryonic development, revealing how different cell types emerge over time. Yet a major question remains: how does this developmental gene activity relate to organismal phenotypes? Here, we introduce the $\mathcal{D}$evelopment-to-$\mathcal{P}$henotype ($\mathcal{D}$–$\mathcal{P}$) rule, which posits that genes with similar developmental expression

**Data availability statement:** Github: https://github.com/ali4lou/D-P-foundations. Zenodo: https://zenodo.org/records/14629057.

**Funding:** This work was supported by grants PID2022-140017OB-C22 (M.C.), PID2019-106116RB-I00 (J.F.P.), and PID2023-151289NB-I00 (J.F.P.) from the Spanish Ministerio de Ciencia e Innovación/Agencia Estatal de Investigación (MICIU/AEI/10.13039/501100011033), and by the European Regional Development Fund (ERDF/EU). A.L. was supported by Ph.D. fellowship PRE2021-099926 from MICIU/AEI/10.13039/501100011033 and the European Social Fund Plus (FSE+). The funders had no role in study design, data collection and analysis, decision to publish, or preparation of the manuscript.

**Competing interests:** The authors have declared that no competing interests exist.

patterns tend to produce similar phenotypic outcomes. Using *Caenorhabditis elegans*, we integrate a detailed single-cell developmental atlas with a structured phenotype ontology to quantify this relationship. We define developmental and phenotypic similarity scores for thousands of genes and find a positive correlation between them, supporting the rule. We also investigate exceptions, such as developmental divergence and phenotypic degeneracy, and identify specific cell types that disproportionately influence phenotypic outcomes. Our work provides a quantitative framework linking gene expression during development to organismal traits.

## Introduction

Cutting-edge sequencing technologies now enable precise tracking of gene activity at single-cell resolution during embryonic development, surpassing earlier methods that could only assess embryos in bulk, e.g., [1]. This advancement is driving the creation of cell atlases –comprehensive maps of cell states and types– providing new insights into developmental dynamics across a wide range of organisms, including zebrafish [2], western claw-toed frog [3], nematode [4], fruit fly [5], or mouse [6].

With the availability of these atlases, numerous issues can now be addressed. For example, they can be applied to better understand the relationship between gene expression, cellular functions, and cell differentiation [7,8], or to provide a comprehensive gene expression map within organoids, facilitating validation against *in vivo* counterparts [9]. At the same time, these data enable a reexamination of classical questions in developmental biology through a quantitative and testable lens, such as uncovering the enhancer logic underlying cell identity [10], determining the proper approaches to defining and organizing cell states and types [11], and exploring the role of cellular plasticity in developmental robustness [12], among others.

Our study contributes to this growing effort by focusing on a well-recognized but rarely quantified principle: the relationship between developmental gene expression patterns and organismal phenotypes. Although it is well established that gene activity during development shapes biological form and function [13], development itself can both constrain [14] and diversify outcomes [15]. What remains lacking is a systematic, cell-resolution framework to rigorously evaluate this relationship. To address this, we introduce the $\mathcal{D}$evelopment–to–$\mathcal{P}$henotype, or $\mathcal{D}$–$\mathcal{P}$, rule, which asks: do genes with similar expression patterns during development tend to result in similar phenotypes? This hypothesis has only recently begun to be evaluated at scale using bulk developmental transcriptomes [16] and emerging approaches such as single-cell phylotranscriptomics [17]. While it necessarily simplifies the complexity of development, it offers a testable framework for quantifying how gene activity maps onto phenotypic outcomes.

However, evaluating this rule presents several challenges. First, accurately computing the developmental details within cellular atlases requires integrating how genes are activated or silenced at specific stages and in particular cell types. Second, while *cellular* phenotypes can be obtained on a relatively large scale,

e.g., [7], capturing *organismal* phenotypes poses a greater difficulty. These two obstacles are addressed using *Caenorhabditis elegans* as a model system.

The first goal is to formalize the developmental information encoded in cell atlases that map the organism's development. To achieve this, a unified developmental signature is proposed, integrating both embryonic time and cell type, thereby enabling the characterization of each gene's activity within this context (Fig 1A, left). This effort utilizes a recent dataset where most cells have been annotated with a cell type or lineage across stages of embryogenesis, primarily from mid-gastrulation to terminal differentiation [4]. Our approach will focus exclusively on annotated cell types. For the second challenge, the Worm Phenotype Ontology (WPO) –a hierarchically structured, controlled vocabulary for standardizing phenotypes [18]– is employed to represent a phenotypic space where each gene is associated with a defined subset of phenotypes (Fig 1A, right). Intrinsic redundancies in the ontology are addressed using non-negative matrix factorization (NMF) [19].

Armed with these datasets, we aim to address three primary questions (Fig A in S1 Text). First, we introduce an *average* developmental and phenotypic similarity score for each gene to quantify how closely its developmental or phenotypic patterns resemble those of others. Genes with higher developmental similarity also tend to have higher phenotypic similarity, consistent with the $\mathcal{D}-\mathcal{P}$ rule. However, some genes deviate from this pattern, prompting us to investigate the mechanisms underlying both phenotypic degeneracy [20] and developmental divergence [21]. Next, we examine how varying levels of developmental specificity correlate with a gene's impact on phenotypes, i.e., pleiotropy [22]. This we do by distinguishing a set of pleiotropic genes. We also identify specific developmental "coordinates" where pleiotropic genes are dominantly expressed. These genes show enrichment particularly at early stages. Finally, we explore whether certain cell types act as key mediators of phenotypic outcomes, referring to this as a *coarse* interpretation of the rule. This work demonstrates how integrating cell atlases and phenotypic ontologies can uncover statistical relationships suggesting how developmental processes may shape phenotypic outcomes, while also highlighting the role of cells as mediators in the genotype-to-phenotype (GP) map.

## Results

### A simple rule within the genotype-phenotype map

To evaluate whether similarity in developmental gene expression trajectories is associated with a higher likelihood of shared organismal phenotypes, a trend we refer to as the $\mathcal{D}-\mathcal{P}$ rule, we construct two quantitative spaces: a developmental space and a phenotypic space. The developmental space reflects the distribution of gene expression across cell types (S1 Table) and embryonic stages, while the phenotypic space captures the impact of gene deactivation on organismal phenotypes (Fig 1A). Each gene is assigned a pair of vectors, one from each space, enabling direct comparisons between developmental patterns and phenotypic outcomes (Methods; extended analysis and Figs B–G in S1 Text); $\vec{g}_D^i$ represents the fraction of cells sampled at a given developmental coordinate in which the corresponding gene *i* is active, while $\vec{g}_P^i$ quantifies the gene *i*'s contribution to an NMF-derived phenotype space.

Using this information, we compute two similarity matrices –one for development and another for phenotype– where each entry represents the pairwise similarity between any two genes in the respective space (Methods; S2 and S3 Tables include similarity values and gene ontology (GO) analysis as functions of these similarities). We then simplified these matrices by computing an average developmental similarity $\langle sim_D \rangle$ and phenotypic similarity $\langle sim_P \rangle$ for each gene, which represent the median of all its associated pairwise values. If the $\mathcal{D}-\mathcal{P}$ rule holds, one should observe a proportional relationship between these two measures. We confirm this relationship (Fig 1B and Figs H and I in S1 Text).

To further examine the robustness of this trend, genes were categorized into distinct ranges of the corresponding $\langle sim_D \rangle$ and $\langle sim_P \rangle$ distributions (below the 20th percentile, between the 20th and 40th percentiles, and so on). The frequency of the relevant phenotypic categories for each developmental class was computed. Under a *strict* interpretation of the $\mathcal{D}-\mathcal{P}$ rule, we would expect phenotypic categories to map one-to-one with developmental expression patterns. We

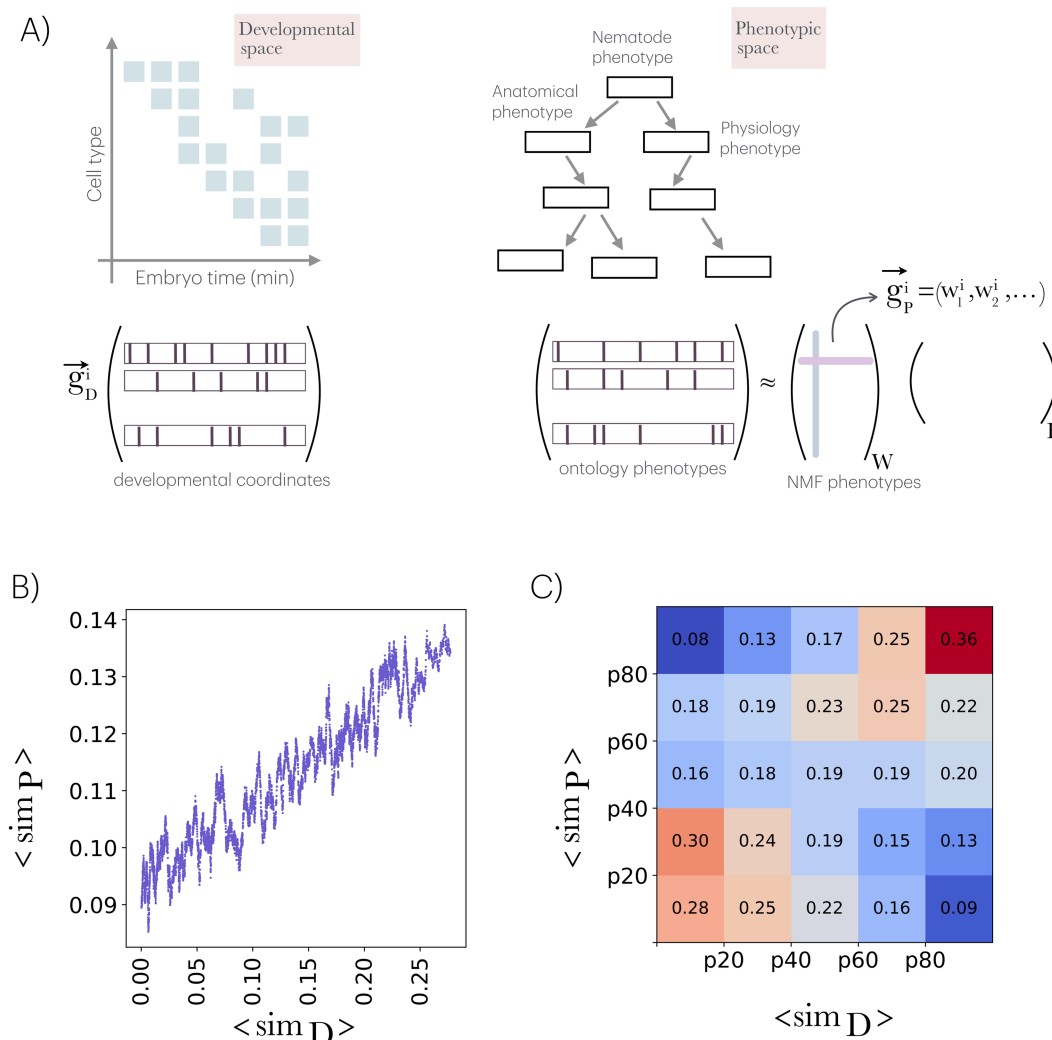

**Fig 1**. **Developmental and phenotypic spaces and the $\mathcal{D}$–$\mathcal{P}$ rule.** A) Developmental and phenotypic spaces are characterized using data from the single-cell atlas and the worm ontology, respectively. For each gene $i$, the relative number of cells in which it is active within the sampled set at each specific developmental coordinate (cell type and embryonic time) constitutes the elements of its associated developmental vector, $\vec{g}_D^i$. Collectively, these vectors define the developmental matrix. Similarly, a phenotypic (binary) matrix is constructed by associating genes with their linked phenotypes in the ontology. This matrix is decomposed using nonnegative matrix factorization, NMF (into a *features* matrix W and a *coefficients* matrix H). In this decomposition, the gene phenotypic vector $\vec{g}_P^i$ is represented by the rows of W (highlighted in pink), while each column (in blue) indicates the contribution of each gene to a specific NMF-derived phenotype. B) By defining and average (median) similarity score for each gene with respect to the rest, we observed a proportionality between $\langle sim_D \rangle$ and $\langle sim_P \rangle$ what highlights the $\mathcal{D}$–$\mathcal{P}$ rule: "similar developmental trajectories lead to similar phenotypes". Plot is a sliding window of the full data (windows size = 100). C) For each set of genes with an average similarity $\langle sim_D \rangle$ within specific percentiles (e.g., p20 indicating less or equal to the 20th percentile, and so on), the percentage of genes belonging to different phenotypic percentile categories is presented (each column sums to 1). Deviations from the expected percentage in development and phenotype suggest a departure from the rule. See main text for further details.

do observe a tendency for genes to be part of equivalent classes; however, we also noted deviations, with some genes showing equivalent developmental patterns but differing phenotypic ones, and vice versa (Fig 1C). These deviations will be studied next.

## Exceptions to the $\mathcal{D}$–$\mathcal{P}$ rule

To analyze discrepancies in the rule, we first fit $\langle sim_P \rangle$ against $\langle sim_D \rangle$ using Loess regression and calculated the residuals for each data point, i.e., deviations from the fit (Figs J and K in S1 Text). This allowed us to define a group that follows the rule with high $\langle sim_D \rangle$ and $\langle sim_P \rangle$ (that we termed D-P genes) and identify two main deviations (D-p and d-P genes, Fig JA in S1 Text).

Fig 2 shows representative gene expression patterns for each class, with $\vec{g}_D$ simplified by averaging across cell types at each embryonic time point (Methods). Developmental *divergent* genes (D-p, which are developmentally similar but phenotypically distinct) are expressed across most cell types and embryonic stages, similar to D-P genes (Fig JB in S1 Text). However, their phenotypic roles differ: while D-P genes are associated with systemic and general phenotypes, D-p genes are linked to a specific subset of enriched phenotypes (S4 Table). This distinction suggests that D-P genes act

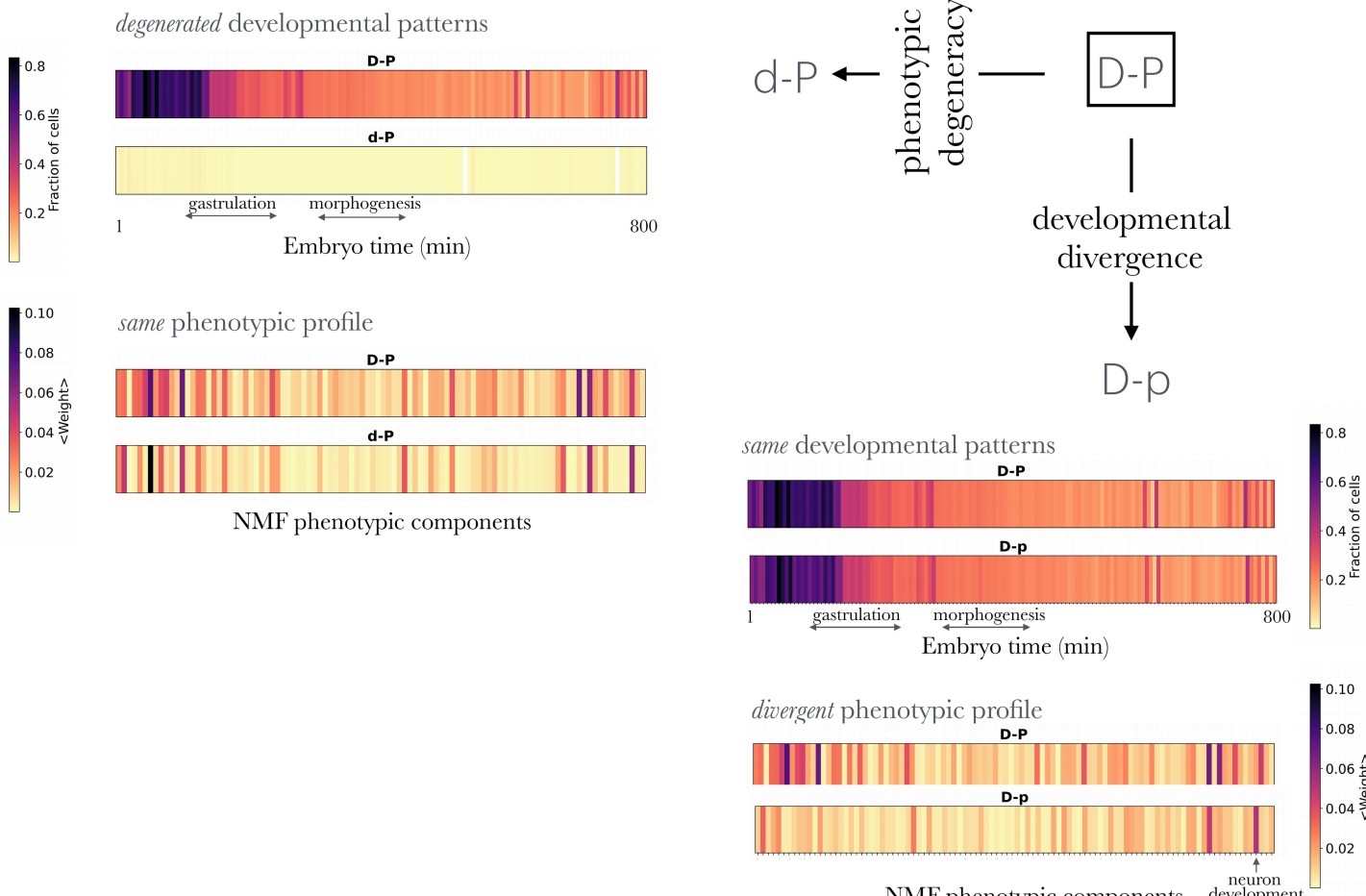

**Fig 2**. **Deviation from the $\mathcal{D}$–$\mathcal{P}$ rule.** We consider the subclass of genes that fulfill the $\mathcal{D}$–$\mathcal{P}$ rule and show strong similarity, denoted D-P, to examine two deviations: phenotypic *degeneracy* or developmental *divergence*. In the case of divergence, denoted as D-p, deviations occur when genes show similar developmental patterns to the D-P class but differ in their phenotypic outcomes (bottom right of the figure). These patterns are represented as barcodes, reflecting the characteristic gene expression of a gene within the corresponding class (averaged across the cell types at each embryonic stage) or its contribution to phenotypic components. In the case of degeneracy, denoted as d-P, deviations occur when genes display different developmental patterns from the D-P class but result in similar phenotypes (top left of the figure). For details, refer to the main text, Methods and Fig K in S1 Text.

as housekeeping genes essential for fundamental processes across all cell types, with their disruption leading to systemic organismal failure. In contrast, `D-p` genes, although also housekeeping genes, play critical roles only in specific cellular contexts. Their loss does not cause a complete systemic failure but results in localized dysfunctions tied to the cell types where their function is indispensable.

Phenotypic *degeneracy* is characterized by genes with similar phenotypic profiles arising from multiple developmental patterns (`d-P`). Unlike `D-P` genes, which are housekeeping genes critical for the organism's overall viability, `d-P` genes exhibit specific developmental profiles, influencing particular cell types or embryonic stages while their associated phenotypes remain systemic and general. Although `d-P` genes are active in a limited number of cell types, their localized functions are vital for organismal survival, with lethality often being one of their enriched phenotypes. See Fig K in S1 Text for additional details and a GO analysis revealing additional differences between the three groups (S5 Table).

### The $\mathcal{D}$–$\mathcal{P}$ rule and the pleiotropy of complex phenotypes

Since genes that exhibit strong similarities and align with the rule's premise impact many phenotypes, we next aimed to examine this more comprehensively using an NMF-derived pleiotropic score, $\mathbb{P}$. This score quantifies the number of traits influenced by a single gene, identifying pleiotropic genes as those exceeding the 95th percentile of the distribution (Fig L in S1 Text). NMF's ability to isolate fundamental phenotypic elements overcomes limitations of traditional pleiotropy measures, which are often constrained by the complexity and interdependence of phenotypes [22,23] (Fig M in S1 Text).

We began by examining the expression patterns of pleiotropic genes. For each gene, we calculated the fraction of sampled cells in which it is expressed (regardless of cell type or embryonic stage), then averaged these values across gene sets. Pleiotropic genes are expressed in significantly more cells than expected by chance. In contrast, non-pleiotropic genes (those below the 5th percentile of the distribution; Fig 3A) are active in far fewer cells and show no notable enrichment for phenotypes or GO terms. Pleiotropic genes, by contrast, are strongly associated with systemic phenotypes and broad functional categories (S6 and S7 Tables).

To incorporate developmental context, we extended the analysis in two ways: by assessing enrichment across embryonic times and by considering the full (cell type, embryonic time) coordinates (Fig 1A). We first computed both the range of embryonic stages each gene is expressed in, as well as the earliest stage of expression. Pleiotropic genes tend to be expressed earlier and across more stages than non-pleiotropic ones (Fig 3B and Fig N in S1 Text). We then analyzed developmental coordinates (cell type, embryo stage) enriched for pleiotropic genes by calculating a *z*-score for expression likelihood at each coordinate (Fig ND in S1 Text). Fig 3C shows the top 100 coordinates enriched for these genes. There exists again a tendency for pleiotropic genes to be enriched in cell types identified at early developmental stages. Note also a subgroup connected to specific intestine and pharyngeal cell types (lower right in Fig 3C); both classes appearing relatively late during development.

Moreover, to examine more explicitly the relationship between the $\mathcal{D}$–$\mathcal{P}$ rule and pleiotropy, we compared the pleiotropy distributions of the three groups of genes (`D-P`, `D-p` and `d-P`) obtained in the previous section depending on their adherence to the rule (Fig 3D). Genes that are strongly similar and fulfill the rule (`D-P` genes) are the ones with the highest pleiotropy values. Moreover, since high $\langle sim_\mathrm{P} \rangle$ values are typically associated with elevated $\mathbb{P}$ (Fig O in S1 Text), this relationship explains the shift toward higher $\mathbb{P}$ observed in `d-P` genes compared to the `D-p` class.

### A coarse interpretation of the $\mathcal{D}$–$\mathcal{P}$ rule

Finally, we take a coarser yet complementary approach to the $\mathcal{D}$–$\mathcal{P}$ rule by grouping the developmental space based on cell types. This method allows us to investigate whether genes expressed in specific cell types are strongly linked to particular phenotypes (Fig 4A), offering an alternative perspective for exploring development–phenotype associations.

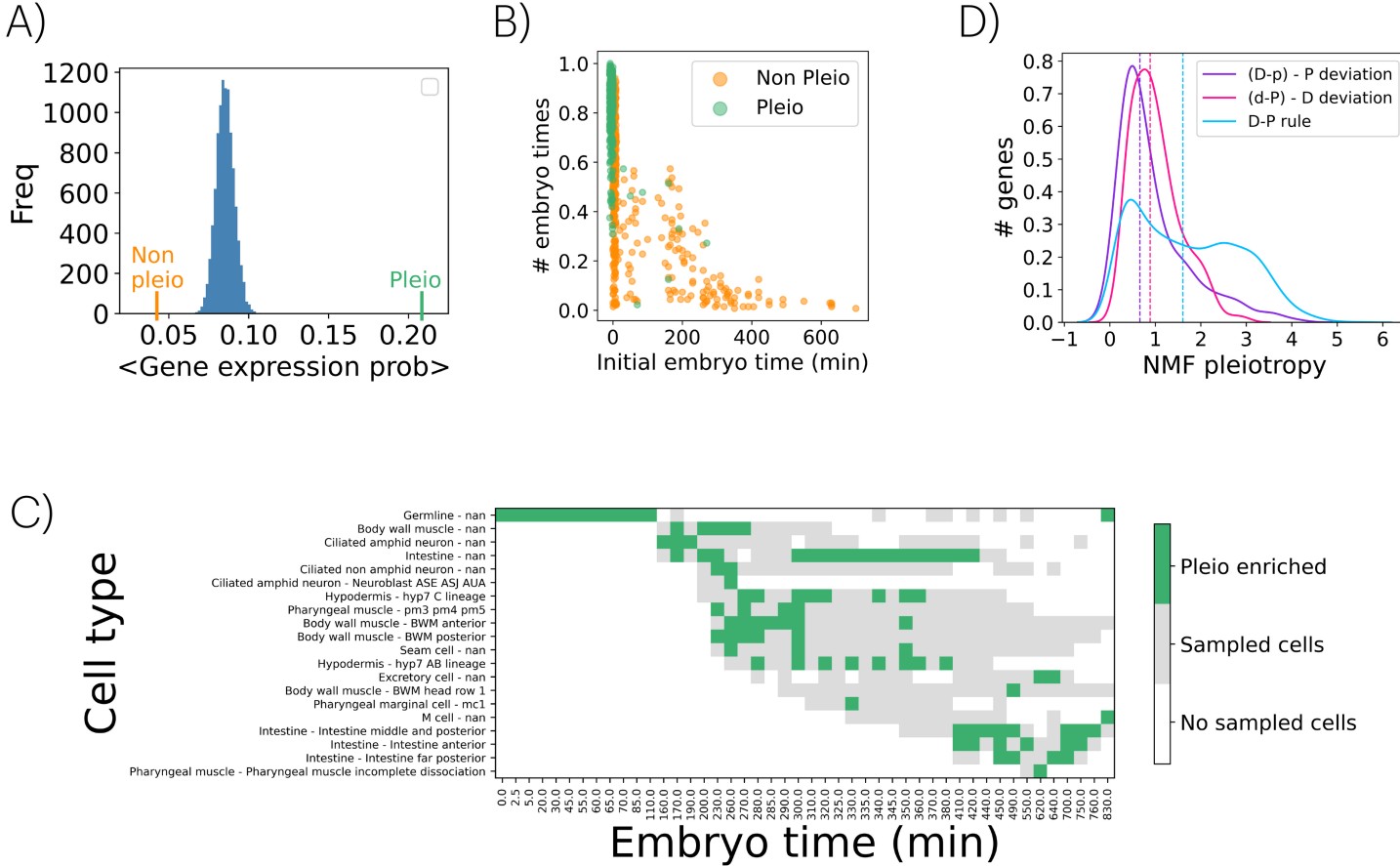

**Fig 3. Pleiotropy and the $\mathcal{D}$–$\mathcal{P}$ rule.** A) Average gene expression probability for non-pleiotropic genes (orange, $n = 418$) and pleiotropic genes (green, $n = 412$). Additionally, the distribution of 10,000 averages of gene expression probabilities, calculated by randomly selecting $n = 415$ genes from the dataset, is shown. B) Each gene of the two sets is shown in an space of initial expression embryo time and fraction of different embryo times in which a genes is expressed. Pleiotropic genes tend to be initially expressed at earlier times in a higher number of stages. C) Distribution of the top 100 pleiotropic genes across developmental coordinates. Enriched regions are highlighted in green, with sampled cell coordinates shown in gray. Note a subgroup associated with specific intestine and pharyngeal cell types, both appearing relatively late in development (lower right). D) Kernel density estimates of pleiotropic score distributions for `D-P`, `D-p`, and `d-P` genes. The $x$-axis represents pleiotropy scores, and the $y$-axis indicates the fraction of genes in each group. Dashed lines show medians, with a mean pleiotropy score of 3.5 for pleiotropic genes.

To attain this, we calculate the fraction of genes expressed in a given cell and the relative contribution of these *active* genes to each NMF-derived phenotype (Methods). The results reveal an expected trend in which cells with a greater number of expressed genes typically display higher cumulative weights for the corresponding phenotype (Fig 4B). However, by performing regression on this trend, we identified cells where the expressed genes have a disproportionately strong influence. If these cells belong to a specific cell type, that type can be identified as a "mediator" of the phenotype using a Kolmogorov-Smirnov (KS) statistical test applied to the distribution of the corresponding regression residuals (Methods). For example, Fig 4B highlights the association between the ASK ciliated sensory neuron and NMF phenotype #49 (KS statistic = 0.94; see also Fig P in S1 Text for additional examples). This phenotype is linked to WPO terms like chemosensory behavior, chemotaxis, and odorant response, corroborating the known role of ASK neurons in detecting chemical stimuli and supporting the relevance of the associations and the statistical approach.

Fig 4C extends this analysis, illustrating the connections between all NMF-derived phenotypes and cell types. Each cell type is linked to at least one phenotype and each phenotype is connected to at least one cell type (Fig Q in S1 Text).

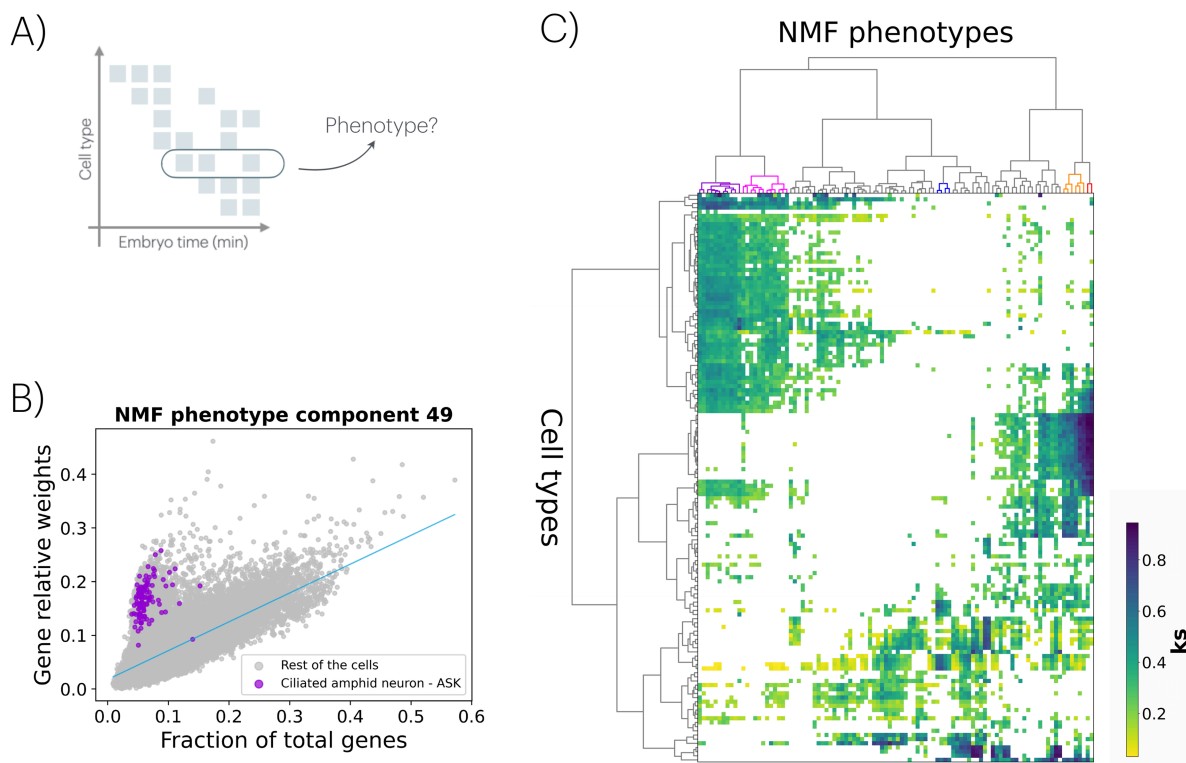

**Fig 4**. **Coarse interpretation of the 𝒟–𝒫 rule.** A) The original developmental space (see Fig 1A) can be simplified by grouping cells by type to assess how strongly these cell types mediate phenotypes. B) By calculating the fraction of expressed genes in each cell (each cell represented as a dot) and the relative contribution of these genes to a specific NMF-derived phenotype (in this case, component #49), we identify a regression pattern. Cells with larger residuals indicate enrichment for the phenotype, with violet dots corresponding to ASK ciliated amphid neurons. This association is determined by comparing the residual distribution of cells from a specific type to that of all other cells using a Kolmogorov-Smirnov (KS) test. Across all cell types, we find a significant association between ASK neurons and phenotype #49 (KS = 0.94, *p*-value < 0.0001). C) By analyzing all NMF phenotypes and identifying cell types with significant KS associations, we generate a heatmap linking NMF phenotypes to cell types. The heatmap's color represents the KS statistic for each association, and all plotted KS values are significant (*p*-value < 0.0001). The *x*-axis lists sorted NMF phenotypes, while the *y*-axis lists cell types, both organized using hierarchical clustering. Dendrograms for each axis are included, grouping related cell types into broader categories, such as precursor and parent cell types, those associated with the nervous system, and others linked to specific anatomical structures (e.g., pharynx, hypodermis, intestine, body wall).

Closer examination of the figure highlights that certain cell types have a stronger influence on complex phenotypes, while some phenotypes are less/more closely associated with specific cell types (S1 Text and S8 Table). Fig 4C also shows distinct cell and phenotype groupings that stand out for having associations with high KS scores: ciliated amphid neurons with chemosensory responses (red in the top dendogram), neurons with acetylcholinesterase inhibitor response (orange), body wall and pharyngeal muscle cells with muscle morphology phenotypes (dark blue), and germline cells with sterile progeny and lethal phenotypes (magenta) or reduced fertility phenotypes (purple). Broader *cell* groupings associated with similar phenotypes are also notable (Fig RA in S1 Text), including precursor and parent cells, nervous system-associated cells, and cells linked to specific anatomical structures (e.g., pharynx, hypodermis, intestine, body wall, S9 Table), with enrichment of these cell types in the *C. elegans* AB lineage (Fig RB in S1 Text).

## Discussion

It is commonly proposed that equivalent gene expression during embryogenesis among a set of genes suggests their potential involvement in analogous organismal phenotypes [21]. However, quantitatively testing this hypothesis has been

challenging. For instance, we lacked single-cell resolution gene expression measurements during development, and large-scale quantification of organismal phenotypes was similarly constrained. By integrating cell atlases and phenotypic ontologies, we aim to overcome these limitations. Importantly, we also consider that embryonic gene expression can influence phenotypes that manifest post-embryonically, as it lays the foundation for later cellular and tissue functions.

One approach to leveraging these datasets is by defining a developmental vector and a phenotypic vector for each gene. This quantification allows us to introduce similarity measures –specifically, to assess how similarly any two genes are expressed during development or how similarly they are linked to phenotypes when perturbed. The $\mathcal{D}$–$\mathcal{P}$ rule hypothesizes proportionality between developmental and phenotypic similarity –a pattern our analysis supports on average– while not overlooking the inherent complexity of phenotypic emergence (Fig 1). Our results indicate that "housekeeping" activity (i.e., ubiquous expression during development) correlates with broader involvement in common phenotypes, while more specific expression profiles align with specialized phenotypes.

There are also deviations from this rule, two of which we examined (Fig 2). First, genes with high expression similarity across many cell types and developmental stages can exhibit diverse phenotypes (`D-P` vs. `D-p`; recall that capitals denote strong similarity, while lowercase indicates dissimilarity; thus, `D-P` genes exhibit strong developmental and phenotypic similarity, adhering to the rule). Despite their broad expression, `D-p` genes are associated with particular phenotypes, explaining their lower $\langle sim_P \rangle$ values. A second deviation involves phenotypic degeneracy (`D-P` vs. `d-P`). In this case, the expression of certain `d-P` genes at distinct developmental stages can drive broad systemic phenotypic changes. These genes are enriched in only a small number of GO terms, reflecting a general heterogeneity of specific cellular functions crucial for organismal viability, such as nucleosome and chromatin functions (S1 Text, S5 Table). These outlier patterns suggest interesting avenues for future experimental follow-up.

That genes broadly expressed or involved in diverse developmental processes might contribute to a wide range of phenotypic effects leads directly to the concept of pleiotropy [12,22,23]. To examine this, we introduced a quantitative measure of pleiotropy by summing the scores assigned to each gene in the feature matrix $W$ obtained by NMF [19]. While this metric may partially reflect research bias –since genes studied more extensively tend to have more annotated phenotypes– it does not necessarily preclude biological relevance [24]. Indeed, pleiotropic genes tend to be expressed in a significantly larger fraction of cells than non-pleiotropic ones, often with broader and earlier-stage expression patterns (Fig 3). Notably, our pleiotropy measure correlates with an independent, experimentally derived metric based on cellular phenotypes resulting from gene knockdowns on *C. elegans* chromosome I [12] (Spearman's $\rho = 0.55$, $p = 8.48 \times 10^{-28}$, S1 Text), reinforcing the idea that cells mediate the manifestation of gene functions even when plasticity may obscure such relationships.

These findings corroborate the notion that pleiotropy is linked to genes with foundational roles in early development, which influence diverse downstream processes. Despite their early and broad expression, subsets of pleiotropic genes exhibit late-stage specificity (e.g., intestine and pharyngeal cell types, Fig 3C), indicating that pleiotropy can also manifest in temporally constrained ways. Finally, the highest pleiotropic scores were observed in genes that exhibit strong similarities and fulfill the $\mathcal{D}$–$\mathcal{P}$ rule (`D-P` class), as high phenotypic similarity correlates with increased pleiotropy ($\langle sim_P \rangle$ scales with $\mathbb{P}$).

In exploring the development-phenotype associations, we grouped cells based on their types and examined their contributions to NMF-derived phenotypes. We identified cell types where those genes expressed during development disproportionately influenced the phenotype. For example, most genes activated in ASK neurons are linked to NMF phenotype #49 (Fig 4), which strongly associates with chemosensory behavior and chemotaxis, aligning with its known role in chemical stimulus detection. This coarse approach enabled us to group phenotypes connected to specific cell types and cell groups associated with similar phenotypes (Fig 4C and S1 Text) what supports the role of cells as a "mediator" of the GP map.

While our approach captures a general trend between developmental expression and phenotype, it also comes with inherent limitations. One such limitation is potential bias from enrichment in study-specific phenotypes. To evaluate

this, we re-derived the rule using a subset of genes selected with more systematic criteria and found that the pattern persisted (Fig S in S1 Text). Another limitation arises from the resolution mismatch between datasets: while gene expression is measured during embryogenesis, many phenotypes are characterized at adult stages. To address this, we again used embryonic cellular phenotypes from Xiao et al. [12] and also observed support for the rule (Fig T in S1 Text). Taken together, these results, and the ability of our approach to recover biologically meaningful associations despite dataset heterogeneity, underscore the robustness of the underlying signal.

Beyond dataset-specific limitations, broader biological factors inherent to *C. elegans* development also influence the applicability and interpretation of the $\mathcal{D}-\mathcal{P}$ rule. These include the convergence of different lineages to produce the same cell type and the minimal divergence in gene expression observed until the final two rounds of cell division [4,25]. We assessed the extent of cross-species generalizability through additional analyses in zebrafish (Figs U, V, and W in S1 Text and S10 and S11 Tables). More broadly, the applicability of our simplification for validating the $\mathcal{D}-\mathcal{P}$ rule in other species may depend on when and how transcriptional differences among cells arise during development. Finally, our approach offers a potentially valuable tool for functional genetic profiling, particularly in disease contexts –including rare and poorly understood conditions with unknown genetic bases [26]. Collectively, these efforts contribute to unraveling the developmental underpinnings of phenotypes and advancing our understanding of how developmental processes shape organismal traits.

## Materials and methods

**scRNA-seq data.** We used the developmental cell atlas of *C. elegans* produced by Packer et al. [4]. The corresponding data is available at the Gene Expression Omnibus (GEO) repository (www.ncbi.nlm.nih.gov/geo) under accession code GSE126954. This dataset includes transcriptome information for 20,222 genes across 86,024 single cells, represented as unique molecular identifier (UMI) counts. For each cell, developmental timing was estimated by correlating its transcriptome with a bulk RNA-seq time series [1]. Additionally, the combination of marker genes, lineage assignments, and developmental timing enabled annotation of the majority of cells by cell type or lineage (Fig F in S1 Text).

**Worm Phenotype Ontology data.** We downloaded *C. elegans* phenotypes and their gene associations (version WS290) from the Worm Phenotype Ontology [18] found in the WormBase (www.wormbase.org/). Gene-phenotype associations were derived from various experimental approaches: RNA interference and genetic allele variations. A gene linked to a particular phenotype is inherently associated with all broader phenotypes that subsume it within the ontology (Figs B and C in S1 Text). In this dataset, the absence of an annotated phenotype does not imply lack of function; detection limits, stage specificity, and redundancy all play a role. To address this, we use continuous similarity scores and apply NMF to reduce redundancy and reveal broader phenotypic patterns (S1 Text).

**Developmental and Phenotypic spaces.** We define a *developmental* space based on the identification of cell types across all quantified cells in the dataset. These types are inferred using gene markers and span different embryo stages. Each gene is thus represented as a "developmental" vector, $\vec{g}_D$, indicating the proportion of cells corresponding to a specific coordinate (cell type × embryonic stage) in which the gene is expressed (Fig 1A; further details can be found in Fig G in S1 Text). To define the *phenotypic* space we create the gene-phenotype association matrix. Matrix rows indicate the presence or absence of phenotypes associated with the alteration of a specific gene. Given that ontologies often contain redundant structures, we approximate this matrix using NMF [19]. The resulting phenotypic vector for each gene, $\vec{g}_P$, represents the weights of that gene across the dimensions of the feature matrix, $W$ (Fig 1A; further details can be found in S1 Text –Figs D and E in S1 Text–, which also discusses alternative definitions of these spaces, Fig I in S1 Text). Importantly, these developmental and phenotypic representations are constructed from independently curated data sources, with negligible overlap arising from shared marker usage, minimizing the risk that their association reflects circular annotation rather than biological signal (S1 Text).

**Pairwise similarities.** We obtain the similarity ($sim_D$, and $sim_P$) between pairs of genes as the cosine distance, $sim(\vec{g}_i, \vec{g}_j) = \frac{\vec{g}_i \cdot \vec{g}_j}{\|\vec{g}_i\|_2 \, \|\vec{g}_j\|_2}$. Cosine similarity is used in gene expression analysis when the focus is on expression patterns rather than the magnitude of expression. Two proportional vectors will have a similarity of 1.

**Developmental and phenotypic patterns for `D-P`, `D-p`, and `d-P` gene classes**. For developmental patterns, we calculate the fraction of cell types expressing each gene at each embryonic stage. We then compute the average profile for the genes in each class (`D-P`, etc.). For phenotypic patterns, we average the NMF phenotypic profiles of individual genes within each class.

**Associations between specific cell types and NMF phenotypes.** We analyze the relationship between the fraction of total genes expressed by each single cell and their relative contribution to a specific NMF phenotype. By the latter we mean the sum of the weights in the NMF matrix $W$ corresponding to genes expressed in the individual cell and associated with that specific NMF phenotype. Then, we perform a regression analysis using data from all individual cells, deriving a regression line that captures the overall trend for the given NMF phenotype. The deviations of each cell from this trend are quantified as residuals, calculated as the differences between observed values and those predicted by the regression line. Grouping the cells into subsets belonging to the same cell type, we perform a KS test of the residual distributions of these subsets versus the rest of the cells. Significant ($p$-value<0.0001) and positive KS statistics indicate that the specific NMF phenotype is related to that cell type (recall that this statistic is bounded between 0 –perfect agreement between distributions–, and 1 –maximally different). We perform this procedure for all the NMF phenotypes (100 components). For each analyzed NMF phenotype, we included only cell types with more than 50 labeled cells (137 cell types).

**Kolmogorov-Smirnov test.** Kolmogorov-Smirnov test compares the sample cumulative distribution $F(x)$ against the reference cumulative distribution $G(x)$. The distributions do not need to be parametric. For this we use the `kstest` function of `scipy.stats`, with the parameter 'greater' (one sided test). In this case, the null hypothesis is that $F(x) <= G(x)$ for all $x$; the alternative is that $F(x) > G(x)$ for at least one $x$.

**Gene Ontology enrichment analysis.** We downloaded Gene Ontology (GO) annotations from WormBase (version WS294). We built three gene–GO term association matrices one for each aspect of the ontology (molecular function, cell component and biological process), by expanding direct annotations to ancestor terms in the GO hierarchy. We calculated the enrichment of terms for different subsets of genes identified along the analysis (Fisher's exact test; fisher exact function from `scipy.stats`). The test accounts for the total number of genes in both the entire dataset and each subset. We corrected $p$-values within each GO aspect for multiple testing using False Discovery Rate.

## Supporting information

**S1 Text. Supplementary analyses and figures SA–SW.**
(PDF)

**S1 Table. List of cell types with their broad classification.**
(XLSX)

**S2 Table. Similarity values. Genes with high and low similarity.**
(XLSX)

**S3 Table. Gene ontology enrichment depending on similarity.**
(XLSX)

**S4 Table. List of `D-P`, `D-p`, and `d-P` genes and corresponding enriched phenotypes.**
(XLSX)

**S5 Table. `D-P`, `D-p`, and `d-P` genes and corresponding GO enrichments.**
(XLSX)

**S6 Table. List of genes with its pleiotropy value.** Pleiotropic and non-pleiotropic genes. Enriched phenotypes for pleio and non-pleio genes.
(XLSX)

**S7 Table. GO enrichment phenotypes for pleio and non-pleio genes.**
(XLSX)

**S8 Table. List of extreme and significant KS values for both cell types and NMF phenotypes associated to Fig 4C.**
(XLSX)

**S9 Table. List of cell types sorted by the hierarchical clustering.** This is linked to the dendrogram of Fig 4C. Each cell type belongs to a specific cluster. These cell types have at least 50 single cells and show a significant KS to a NMF-based phenotypic component.
(XLSX)

**S10 Table. List of `D-P`, `D-p`, and `d-P` genes and corresponding enriched phenotypes for zebrafish.**
(XLSX)

**S11 Table. Further analysis of zebrafish (S1 Text).** List of the most influential original phenotypes in each NMF cluster of Fig WA in S1 Text. List of extreme and significant KS values for both cell types and NMF phenotypes associated with Fig WA in S1 Text.
(XLSX)

## Author contributions

**Conceptualization:** Mónica Chagoyen, Juan F. Poyatos.

**Data curation:** Alicia Lou.

**Formal analysis:** Alicia Lou, Juan F. Poyatos.

**Funding acquisition:** Mónica Chagoyen, Juan F. Poyatos.

**Investigation:** Alicia Lou, Mónica Chagoyen, Juan F. Poyatos.

**Project administration:** Juan F. Poyatos.

**Software:** Alicia Lou.

**Supervision:** Juan F. Poyatos.

**Writing – original draft:** Juan F. Poyatos.

**Writing – review & editing:** Alicia Lou, Mónica Chagoyen, Juan F. Poyatos.

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
