## [Decision Letter · Decision Letter 0]

18 Jun 2025

PCOMPBIOL-D-25-00236

Cell atlases and the Developmental Foundations of the Phenotype

PLOS Computational Biology

Dear Dr. Poyatos,

Thank you for submitting your manuscript to PLOS Computational Biology. After careful consideration, we feel that it has merit but does not fully meet PLOS Computational Biology's publication criteria as it currently stands. Therefore, we invite you to submit a revised version of the manuscript that addresses the points raised during the review process.

Please submit your revised manuscript within 60 days Aug 18 2025 11:59PM. If you will need more time than this to complete your revisions, please reply to this message or contact the journal office at ploscompbiol@plos.org. Please include the following items when submitting your revised manuscript:

We look forward to receiving your revised manuscript.

Kind regards,

Dieter Vanderelst

Academic Editor

PLOS Computational Biology

Marc Birtwistle

Section Editor

PLOS Computational Biology

**Additional Editor Comments :**

I would like to thank the authors for their patience during the handling of this manuscript. Challenges in obtaining timely reviews, combined with my own scheduling conflicts, led to a review process that took longer than usual. Typically, PLoS Computational Biology requires three reviews; however, due to the already extended review timeline and the quality of the two reviews we received, I decided to proceed with two reviews.

Both reviewers agree that the manuscript's central question is interesting and worth exploring. However, Reviewer 1 is critical, suggesting that the relationship examined in the paper is already well understood in the field and that the manuscript does not clearly articulate its conceptual or empirical novelty. Nonetheless, both reviewers acknowledge that the work is technically competent and methodologically sound.

Reviewer 2 recommends revising the manuscript, noting that the analyses are intriguing and the findings are likely to be valuable, especially if issues related to presentation and interpretability are addressed—such as clearer figures and stronger framing.

Reviewer 1 raises more significant concerns about biological complexity, data resolution mismatches, the lack of experimental validation, and potentially overstated conclusions. These criticisms are substantial and should be addressed, but they do not necessarily undermine the core analytic approach or the premise of the D–P rule as a testable framework. Addressing these concerns could take the form of either expanding the work, where possible or clarifying and reframing the questions addressed and the implications of the paper.

With the appropriate revision, the manuscript has the potential to make a meaningful contribution to our understanding of the genotype-phenotype map. In particular, the idea that developmental expression context — independent of molecular function — may, in some cases, serve as a strong predictor of phenotypic outcome is both conceptually and practically significant. Properly framed, the manuscript provides a valuable counterpoint to protein-centric models of functional prediction, demonstrating that shared developmental contexts can yield convergent phenotypic consequences, even across genes with distinct molecular roles.

I therefore recommend a major revision.

**Journal Requirements:**

4) Please ensure that the funders and grant numbers match between the Financial Disclosure field and the Funding Information tab in your submission form. Note that the funders must be provided in the same order in both places as well. Currently, the order of this grant "PRE2021-099926" is different in both places. In addition, "ERDF/EU” and FSE" are missing from the Funding Information tab.

5) Please provide a completed 'Competing Interests' statement, including any COIs declared by your co-authors. If you have no competing interests to declare, please state "The authors have declared that no competing interests exist".

6) Thank you for stating "Github: https://github.com/ali4lou/D-P-foundations Zenodo: https://zenodo.org/records/14629057." Please ensure to provide a complete Data Availability Statement in the online submission form.

**Reviewers' comments:**

Reviewer's Responses to Questions

Reviewer #1: Summary

This manuscript introduces a conceptual framework termed the "Development-to-Phenotype (D–P) rule," which aims to quantify the relationship between developmental gene expression and organismal phenotypes using C. elegans single-cell transcriptomic data and the Worm Phenotype Ontology (WPO). The authors propose that genes with similar embryonic expression patterns tend to result in similar phenotypes, and they explore exceptions to this trend as well as links to pleiotropy.

Comments

1. Lack of conceptual novelty and predictable findings

The central idea that developmental gene expression correlates with phenotypic outcomes is well established in developmental biology and systems genetics. While this study provides a formal framework to quantify that relationship, the main conclusions are largely expected. Previous studies have explored similar genotype–phenotype relationships using expression atlases and phenotype ontologies. The manuscript does not sufficiently distinguish its contribution from this existing body of work. Moreover, many results, such as the observation that broadly expressed "housekeeping" genes tend to be pleiotropic and associated with systemic phenotypes, are intuitive and consistent with known biology. Although the analysis is technically sound and clearly presented, the findings are largely confirmatory and lack mechanistic novelty.

2. Oversimplification of phenotype emergence

The D–P rule rests on the assumption that similar developmental expression patterns should lead to similar phenotypic outcomes. This overlooks the inherent complexity of how phenotypes arise. Phenotypes are often emergent, context-dependent, and shaped by subtle primary defects that manifest as divergent secondary traits. In many cases, so-called "phenotypes" are higher-order or secondary outcomes that arise from cascades of subtle primary molecular or cellular defects. Two genes may cause seemingly similar organismal phenotypes (e.g., uncoordinated movement) through entirely different mechanisms and cell types. Conversely, two genes with very similar roles in development may lead to divergent or even undetectable phenotypes depending on buffering mechanisms, measurement resolution, or stage-specific compensation.

Additionally, gene redundancy and network buffering may prevent any observable phenotype from emerging, even when a gene has important developmental roles. A gene with specific and important developmental expression may not produce any overt phenotype when perturbed, due to functional compensation by paralogs or pathway-level redundancy. These "no phenotype" cases are especially problematic in datasets like the WPO, where negative results are underrepresented and often not systematically reported.

These biological realities make it unreasonable to expect consistent phenotypic similarity based solely on expression profiles. The authors should address these limitations more explicitly, and consider reframing the D–P rule as a probabilistic tendency rather than a deterministic rule. Incorporating measures of phenotype specificity or annotation confidence could help mitigate these issues.

3. Resolution mismatch between datasets

There is a fundamental mismatch between the resolution of the gene expression and phenotype data. The expression data are derived from a single high-resolution embryonic single-cell RNA-seq dataset, while the WPO phenotypes are compiled from heterogeneous sources spanning multiple life stages and experimental conditions. This discrepancy introduces substantial noise. Genes with broad phenotype annotations may reflect research bias rather than biological pleiotropy, while genes with few annotations may be under-studied rather than functionally narrow.

To strengthen the analysis, the authors should stratify phenotypes by source or method (e.g., curated vs. high-throughput RNAi), normalize for annotation density, and consider focusing on more standardized datasets, such as genome-wide RNAi screens. Benchmarking against a high-confidence gene-phenotype subset could also help validate key conclusions.

4. Temporal misalignment between expression and phenotype

The study uses embryonic gene expression data but compares it to phenotype annotations that include post-embryonic and adult stages. Many of the phenotypes analyzed likely arise from gene functions outside the developmental time window studied, weakening the claim that embryonic expression patterns alone predict phenotypic outcomes. The authors should consider restricting phenotype analysis to embryonic or early post-embryonic phenotypes, or at least stratifying phenotypes by developmental timing where possible. This limitation should also be discussed more explicitly in the manuscript.

5. Lack of experimental validation

The entire study is computational and retrospective. Given the central hypothesis, experimental validation would add significant value. For example, the authors could use their similarity matrices to predict phenotypes of understudied genes and test a small number through RNAi perturbation, followed by phenotypic assays, ideally during embryogenesis, to match the expression data. Even partial validation of such predictions would greatly strengthen the paper and demonstrate that the D–P rule has practical utility beyond descriptive modeling.

6. Unvalidated cell-type–phenotype associations

The identification of specific cell types such as ASK neurons as key mediators of certain phenotypes (e.g., chemotaxis) is an interesting aspect of the analysis. However, this claim remains correlative. The authors could bolster this part of the study by perturbing genes enriched in these cell types and assessing the predicted behavioral outcomes. Cell-type–specific rescue experiments could also help establish causality.

7. Independent, matched-resolution validation

The manuscript would benefit from independent validation using a dataset that aligns more closely in both temporal and spatial resolution with the embryonic single-cell transcriptomic data used in this study. While the authors rely entirely on the WPO, which is heterogeneous and coarse-grained, they cite, but do not incorporate, a more appropriate resource: the study by Xiao et al. (2022), which provides systematically measured embryonic phenotypes at single-cell resolution following gene perturbation. This dataset offers a rare opportunity to validate the D–P rule under more controlled and resolution-matched conditions. Using it to test whether genes with similar developmental expression profiles also yield similar embryonic phenotypes would directly support (or challenge) the study’s central claim. At minimum, comparison of trends between the WPO-based analysis and those derived from the Xiao dataset would clarify the generalizability of the findings.

8. Overstatement of conclusions

At times, the manuscript overstates the strength of its findings. The observed correlation between expression and phenotype similarity is only moderate and largely average-case. Interpretations of exceptions to the rule (e.g., D-p or d-P genes) remain speculative without empirical follow-up. The conclusions would benefit from a more cautious and nuanced tone.

Recommendation

I do not recommend publication in PLOS Computational Biology in its current form. The work is carefully executed but primarily descriptive, with limited novelty and no experimental validation. The conceptual framework oversimplifies phenotype emergence and overlooks key biological complexities. If the authors can significantly revise the manuscript to address these concerns, particularly by clarifying the novelty, validating key predictions, and aligning datasets more closely, then the study may be better suited for a venue focused on computational analysis or integrative genomics.

Reviewer #2: This is a solid paper with a number of interesting analyses and results. I recommend a revise and resubmit, with a favorable opinion of acceptance. There are a number of issue that need to be addressed beforehand, which I have presented below:

First paragraph on Page 3: this is the most important paragraph of the Introduction. I think I follow the flow and logic, but a schematic showing the different questions and associated metrics would improve comprehensibility.

* I assume that Figure 1A is a version of what I am requesting for a new Introductory Figure (#1). If so, please use this as the basis for the new Figure 1.

A few words about the D-P rule:

* this definition would probably only hold true for C. elegans.

* as a general phenomenon, you are essentially talking about developmental buffering. If would be nice to talk a bit more about the cell-specific mechanisms you propose for each version of D-P, perhaps in the context of the lineage tree. I know that you mention the C. elegans lineage tree in passing, but this needs a more in-depth discussion.

Figures in Results:

Figure 2: interesting approach in using barcodes to provide intepretability to the different cases of convergence and divergence in development. Yet it might also help to annotate the figure a bit more so that the patters we are looking for are made explicit.

* I am also not clear on how to interpret Figure 3. For the most part, this figure is difficult to interpret, but Figure 3C helps to summarize the significance of the analysis.

* Figure 4, or the coarse interpretation of the D-P rule, is the least convincing part of this analysis. The "courseness" here refers to the cell types. It does seem like the deterministic cell fates of C. elegans will make these results a bit less generalizable.

I like the "intrinsic features" section of the Discussion, which highlights the potential issues with generalizing from C. elegans development. But you do need to speculate on how this D-P rule will vary or even be applicable across species.

Good data integration overall (scRNA-seq + annotation), and the methods look sufficient. Data is made available in a Zenodo repository. The code availability is also acceptable (Github repository).

**Have the authors made all data and (if applicable) computational code underlying the findings in their manuscript fully available?**

Reviewer #1: Yes

Reviewer #2: Yes

PLOS authors have the option to publish the peer review history of their article (what does this mean?). If published, this will include your full peer review and any attached files.

Reviewer #1: No

Reviewer #2: **Yes:** Bradly Alicea

**Figure resubmission:**
---

## [Decision Letter · Decision Letter 1]

23 Dec 2025

PCOMPBIOL-D-25-00236R1

Cell atlases and the Developmental Foundations of the Phenotype

PLOS Computational Biology

Dear Dr. Poyatos,

Thank you for submitting your manuscript to PLOS Computational Biology. After careful consideration, we feel that it has merit but does not fully meet PLOS Computational Biology's publication criteria as it currently stands. Therefore, we invite you to submit a revised version of the manuscript that addresses the points raised during the review process.

We look forward to receiving your revised manuscript.

Kind regards,

Dieter Vanderelst

Academic Editor

PLOS Computational Biology

Marc Birtwistle

Section Editor

PLOS Computational Biology

**Additional Editor Comments:**

Dear authors

Thank you again for your patience during this protracted review process.

We have now received an additional review (Reviewer 3). This reviewer raises a single but interesting and legitimate point. At the same time, I sense that the issue they raise is not entirely "resolvable" in a strict sense and may be inherent to the kinds of data and annotations that are available in this domain: while one could argue that shared conceptual structure in developmental and phenotypic annotations might inflate correlations, the converse is also plausible. Variability, incompleteness, and heterogeneity in phenotype annotations could attenuate the observed signal. In that sense, the reported correlations may represent a lower bound rather than an artifact.

Given this, I would like to ask you to acknowledge or discuss the reviewer's issue and clarify the interpretive limits of the analyses, including factors that could reduce the magnitude of their findings. Once you have completed this, I will proceed to accept the paper. No formal reply to the reviewer is needed: if you can resubmit the paper with a brief comment on what has changed, I will make a final decision without an additional round of reviewing.

Thanks again for your patience.

Best

Dieter

**Journal Requirements:**

1) We have noticed that you have uploaded Supporting Information files, but you have not included a complete list of legends. Please add a full list of legends for your Supporting Information files after the references list.

2) Please ensure that the funders and grant numbers match between the Financial Disclosure field and the Funding Information tab in your submission form. Note that the funders must be provided in the same order in both places as well.

**Reviewers' comments:**

Reviewer's Responses to Questions

**Comments to the Authors:**

Reviewer #2: The authors have addressed the points raised in Round 1 to my satisfaction.

Reviewer #3: The authors investigate the statistical relationship between gene expression trajectories during development and subsequent phenotypes. I found this to be an interesting idea and the paper to be well written. I had one fundamental concern/query that I think is critical to the analysis and the authors should address prior to publication.

Their results rely on a statistical analysis of the association between developmental trajectories and phenotypes, using two distinct data sources (scRNA-Seq data for the developmental trajectories Ontology data for the phenotypes). This analysis is essentially correlative and so is at risk of confounders. I think that it must first be clearly established that these two datasets and/or their interpretation of them do not rely implicitly on some shared information (or indeed make use of each other in an opaque way) since the presence of such confounders would undermine their analysis. For example, if the scRNA-Seq annotations use information from WormBase then information could leak from the ontology to the scRNA-Seq data and this could generate positive associations. Similarly, if some of the gene associations in the ontology data indirectly take into account developmental information of some kind (that is in general, not the specific scRNA-Seq data they use) then associations would also be expected but not particularly informative. Such information leakage issues can be quite subtle and hard to spot, particularly when drawing data from numerous sources, each of which have their own complicated analysis pipelines. Could they comment on this and perhaps reassure the reader that such issues are not relevant or have been dealt with appropriately?

**Have the authors made all data and (if applicable) computational code underlying the findings in their manuscript fully available?**

Reviewer #2: Yes

Reviewer #3: Yes

PLOS authors have the option to publish the peer review history of their article (what does this mean?). If published, this will include your full peer review and any attached files.

Reviewer #2: **Yes:** Bradly Alicea

Reviewer #3: No

**Figure resubmission:**
---

## [Editor Report · Decision Letter 2]

25 Jan 2026

Dear Dr. Poyatos,

We are pleased to inform you that your manuscript 'Cell atlases and the Developmental Foundations of the Phenotype' has been provisionally accepted for publication in PLOS Computational Biology.

Best regards,

Dieter Vanderelst

Academic Editor

PLOS Computational Biology

Marc Birtwistle

Section Editor

PLOS Computational Biology

---

## [Editor Report · Acceptance letter]

PCOMPBIOL-D-25-00236R2

Cell atlases and the Developmental Foundations of the Phenotype

Dear Dr Poyatos,

I am pleased to inform you that your manuscript has been formally accepted for publication in PLOS Computational Biology. Your manuscript is now with our production department and you will be notified of the publication date in due course.

With kind regards,

Anita Estes
